# Study on Thermal Performance of Single-Tank Thermal Energy Storage System with Thermocline in Solar Thermal Utilization

Chao Zhu [1], Jian Zhang [2], Yueshe Wang [2,*], Zehong Deng [2], Peng Shi [1], Jian Wu [1] and Zihao Wu [1]

[1] State Grid Shaanxi Electric Power Research Institute, Xi'an 710100, China; zhuchao_xjtu@163.com (C.Z.); sp5970@sina.com (P.S.); wj96012022@163.com (J.W.); wzh15291999899@163.com (Z.W.)
[2] State Key Laboratory of Multiphase Flow in Power Engineering, Xi'an Jiaotong University, Xi'an 710049, China; zhangjian1998@stu.xjtu.edu.cn (J.Z.); dengzh1994@foxmail.com (Z.D.)
* Correspondence: wangys@mail.xjtu.edu.cn

**Abstract:** For the intermittence and instability of solar energy, energy storage can be a good solution in many civil and industrial thermal scenarios. With the advantages of low cost, simple structure, and high efficiency, a single-tank thermal energy storage system is a competitive way of thermal energy storage (TES). In this study, a two-dimensional flow and heat transfer model of a cylindrical storage tank with water as heat transfer fluid (HTF) is developed, in which the effects of time, flow velocity, and height-to-diameter ratio of the tank on the thermocline thickness have been highlighted. The results show that the thermocline thickness in the storage tank is increasing during the charging and discharging processes, and it increases with the increase of the inlet flow velocity and the height to diameter ratio. It is emphasized that in our cases when the time period of $t$ is 14,400 s, the fluid inlet velocity of $u_{in}$ is $4.577 \times 10^{-4}$ m/s, and the height-to-diameter ratio of $H/D$ is 1.2, the performance evaluation index reaches the maximum 0.9575, and the efficiency of the system is the highest. It is expected that all of the findings herein can provide a fundamental understanding of the design and operation of the single-tank thermal energy storage system in water heating for civil and industrial applications.

**Keywords:** single-tank system; thermal energy storage (TES); thermocline; thermal performance





## 1. Introduction

In order to honor the Paris Agreement on climate change, recently, China aims to peak carbon dioxide emissions before 2030 and achieve carbon neutrality before 2060 by adopting more vigorous policies and measures. To achieve these goals, the fraction of new energy, i.e., solar energy and wind energy, in the secondary energy sources such as electric power has to be increased in the replacement of fossil energy in China. However, due to its intermittence and randomness, the high permeability of new energy is unconducive to the safe operation of the whole power grid system. So far, various energy storage technologies are having been adopted as a promising policy and measure to overcome these bottleneck problems because they are able to not only dispatch the discontinuity and inhomogeneity in the spatial and temporal distribution of new energy but also deploy the requirement of frequency and amplitude modulation in the power grid.

Thermal energy storage (TES) is widely employed in the combined heat and power (CHP) power plants, solar energy thermal utilization, and concentrating solar power (CSP) plants. In general, heat transfer fluid (HTF) is charged by the power plant cooling water, solar collector, or solar receiver and flows into a high temperature tank. Moreover, during the discharging HTF is pumped into the heat exchanger to heat transfer to the working fluid and flows into a low temperature tank. This is the working procedure of the two-tank TES system. Up to now, a single-tank thermal energy storage system is becoming a novel TES. As shown in Figure 1, a solar hot water system is based on a single-tank thermal energy storage technology. The system consists of a set of flat plate collectors, a storage

tank, a controller, the user, a charging pump, a feed valve, a supply valve and a supply pump. The flat plate collectors absorb solar radiation and heat the water. When flat plate collectors produce abundant hot water, the controller starts the charging pump and the hot water is stored in the storage tank. When the user needs hot water, the controller opens the supply valve and starts the supply pump, then the user will obtain the hot water. For the single-tank TES, the charged HTF is sprayed from the top through a uniform flow orifice into the tank filled with porous media made of quartz stone, small sand gravel, or other solid heat storage materials. When discharging, high temperature HTF is pumped out from the top of the tank and discharged low temperature HTF is squeezed back from the bottom of the same tank (as shown in Figure 2). Meanwhile, there must be a layer of fluid of HTF, called the thermocline, to isolate the charged fluid and the discharged one. The thinner the thickness of the thermocline is, the higher the efficiency of the TES is. Compared with two-tanks, a single-tank thermal energy storage system has the prominent potential of saving the investment in the first hardware and infrastructure. Obviously, the thermocline thickness is a significant parameter for evaluating the heat performance of the single-tank TES system. Therefore, much research has been done on single-tank thermal energy storage systems, especially on thermocline.

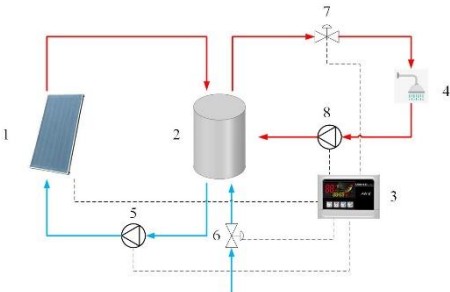

**Figure 1.** Schematic diagram of a solar hot water system (1-flat plate collector, 2-storage tank, 3-controller, 4-user, 5-charging pump, 6-feed valve, 7-supply valve, 8-supply pump).

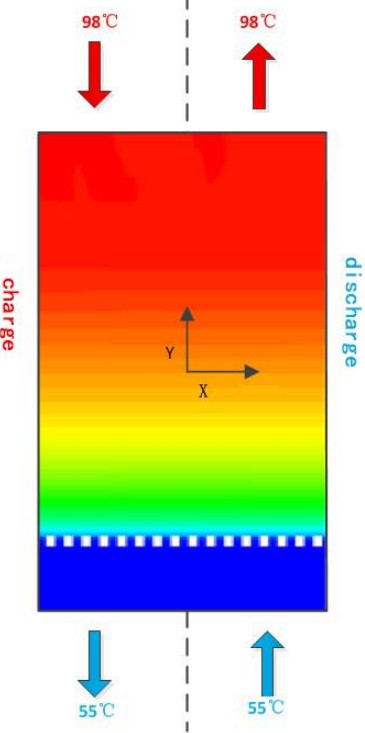

**Figure 2.** Flow and heat transfer diagram of two-dimensional model.

The majority of previous studies on the single TES were based on numerical simulations. Afrin et al. [1] numerically analyzed how the difference in the percentage of porous medium influences the effectiveness of the flow-distribution and hence, the overall performance of the TES system. Tse et al. [2] described a thermodynamic model that simulates the discharge cycle of a single-tank thermal energy storage system that can operate from the two-phase (liquid-vapor) to supercritical regimes for storage fluid temperatures typical of concentrating solar power plants. Based on the second-law of thermodynamics, Zavattoni et al. [3] evaluated the thermal stratification of a single-tank TES system by accurate time-dependent 3D CFD simulations. Abdulla et al. [4] conducted a comprehensive numerical simulation of a 125 MWh$_t$ thermocline tank and examined the effect of relevant design and operating parameters on the performance of the TES system. Capocelli et al. [5] developed a thermophysical model of an innovative thermal energy storage facility, which is ideated, realized, and tested by ENEA (Italy) and tested properly for this particular geometry. The influence of the radial position (r) on the thermocline degradation was studied, finding that at the edges (r→1) the thermocline remains unchanged for longer (around ten times more) than at the center of the tank (r→0). Raccanello et al. [6] analyzed the behavior of the most common single-tank configurations of thermal storage capacities that involve a transfer of mass (open systems) or/and heat (closed/hybrid systems), using simplified dynamic models of different complexity: zero-dimensional, quasi-one-dimensional and one-dimensional. Chang et al. [7] compared analytical approach solutions derived from the one-dimensional two-phase (1D-2P) without heat loss model and one-dimensional one-phase (1D-1P) with heat loss model with experimental data, and found that the local thermal equilibrium assumption is inappropriate in the thermocline tank. Mojtaba et al. [8] proposed a new criterion for quantifying thermocline thickness and defined an ideal charging diffuser. The accuracy of the obtained results was validated against available experimental data and a good agreement was observed. Reddy et al. [9] applied a comprehensive laminar and k-$\varepsilon$ turbulent flow energy transport model to calculate the stability of the thermocline, and found that the Re = 1 provides better stability in the axial and radial direction as well as diagonal than other Reynolds numbers. Parida et al. [10] numerically studied the effects of an inlet inertial jet on the thermal blending of hot and cold heat transfer fluid (molten salt) for a single tank sensible thermal energy storage system. These results revealed that the single tank thermocline storage performance can be improved for higher charging/discharging rates using hemispherical diffusers. Cheng et al. [11] adopted the computational fluid dynamics method to study fluid dynamics and thermal performance of a new ternary molten salt of $KNO_3$-$NaNO_3$-$NaNO_2$ in a heat storage single tank with a thermocline. Cao et al. [12] numerically analyzed the thermocline characteristics of the thermal storage tank during the actual operation. The results showed that the thermocline thickness increases with the operating time and the growth rate gradually slows down. Ge et al. [13] numerically studied the influence of the number and the diameter of openings, the diameter of the water distributor, and the ratio of height to diameter of the tank on the discharging performance of the regenerator. The results showed that the greater the number of openings, the larger the opening diameter, or the larger the hole diameter, the thinner the thermocline. Wang et al. [14] designed a water single-tank thermocline thermal energy storage experimental system. They investigated the process including discharging, dissipation, cyclic operation, and simultaneous charging and discharging by numerical simulation totally. Their results indicated that as the number of cycles increases the thermocline thickness increases continuously. Yin et al. [15] established thermal storage tank models under different working conditions. They recognized that the installation of heat insulation panels can effectively prevent the mixing of hot and cold fluids, thereby improving the heat storage performance of the heat storage tank.

Furthermore, Gaggioli et al. [16] presented the experimental results obtained by ENEA in the Casaccia Research Centre (Rome, Italy), with a small-scale test section consisting of a 300 kW$_{th}$ steam generator inserted in an 8 m$^3$ storage tank with molten salt at high temperature. Gajbhiye et al. [17] conducted an experimental analysis of a single media

thermocline storage system equipped with a perforated, eccentrically mounted, vertical, and porous flow distributor, and analyzed thermocline formation and propagation during the charging cycle. It is observed that the thermocline layer is thinner for higher flow rates. Advaith et al. [18] carried out an experimental investigation of the sparsely studied 'single medium thermocline' (SMT) based single tank sensible thermal energy storage and discussed the effect of Atwood number (density stratification) and mean temperature on TES effectiveness as well as providing the appropriate thermodynamic insights. Yin et al. [19] proposed a kind of hybrid thermocline heat storage method in multi-scale structures and relevant experimental systems and experimentally investigated the fluid flow and heat storage performances of molten salt in a multi-scale structure. The results showed that the theoretical heat storage efficiencies among the three heat storage manners are less than 80% because of the existence of the thermocline. Yang et al. [20] investigated the characteristics of thermal energy storage using the mixture of Xceltherm 600 synthetic oil and sand under various inlet temperatures and velocities. It is found that the TES efficiency of the sand-oil mixture increased by 18.5% compared to that of pure sand. Zhang et al. [21] selected the binary nitrates ($KNO_3$ + $Ca(NO_3)_2$) as a heat storage medium, and comparatively studied three different inlet velocity conditions such as constant value, manual, and automatic adjustment. The results showed that the heat discharging power drops with the heat discharging time under constant inlet velocity.

A lot of research on the single-tank thermal energy storage system adopted molten salt as a storage medium, and the studies based on water were rare. However, in some application scenarios, such as the combined heat and power (CHP) plants and solar heating for civil architecture and industrial application, the single-tank thermal energy storage system based on water is commonly employed because of its low cost and no solidification in a large range of temperature. Additionally, due to the low viscosity of water, it is easy to flow in disturbance to thin the thermocline and improve system efficiency. Therefore, the objective of this study is to attain a comprehensive investigation of the thermal performance of the single-tank thermal energy storage system as water is chosen as the storage medium. The performance of a single-tank TES system is closely related to the thermocline thickness. The thicker the thermocline is, the worse the performance is, so a novel efficiency evaluation model for the single-tank TES technology based on thermocline thickness is proposed in this paper. In order to highlight some crucial factors of thermocline on the thermal performance, a two-dimensional flow and heat transfer model of the single-tank thermal energy storage system is established, and the effects of time, flow velocity, and height-to-diameter ratio on thermocline thickness are considered.

## 2. Modeling

The study focuses on a cylindrical heat storage tank with water as HTF. The geometric modeling is depicted in Figure 1. During the numerical simulation, a kind of structural grid is employed, and the grid independence is verified according to the temperature distribution results.

### 2.1. Physical Model

The influences of three factors are concerned, respectively: time, flow velocity, and height-to-diameter ratio. The range of height-to-diameter ratio is between 0.4 and 1.4, and the variation is 0.1. The flow velocities are 425, 525, 625, and 725 m$^3$/h during the charging process and 850, 1050, 1250, and 1450 m$^3$/h during the discharging, respectively. The variation of temperature distribution with time under different conditions is simulated.

In this paper, the cylindrical storage tank is vertically placed. Before discussing the storage tank, the following assumptions are proposed:

① The fluid flow and heat transfer in the tank are one-dimensional. Under this assumption, the thermocline is axisymmetric regardless of radius. Water is evenly introduced into and led out of the storage tank.

② The insulation of the tank body is extraordinarily good, so the heat loss caused by heat transfer between the tank body and outside is extremely small and can be ignored. Under this assumption, the change of thermocline in the tank is mainly caused by the heat conduction and mixing of cold and hot fluids, but there is nothing to do with the heat conduction of the tank.

③ During the charging process, hot water is injected from the top of the storage tank, and the temperature of the incoming water is higher than that of the water already stored in the tank. During discharging process, cold water is injected from the bottom of the tank. Consequently, the mixing between cold and hot water caused by density differences can be avoided due to different water temperatures.

④ The temperature variation of water in the whole process is large, so it is considered that the physical properties change in flow process.

⑤ The whole process is regarded as unsteady flow to obtain the solution at each time.

Because of assumption①, the three-dimensional model is simplified to two-dimensional. Figure 2 shows the charging mode of the storage tank. The storage medium in the storage tank is water. At this time, the high-temperature water (98 °C) enters from the top of the storage tank at a fixed flow rate. According to Figure 2, the low-temperature water (55 °C) flows out from the bottom of the storage tank. During the discharging process, the direction of the flow is opposite.

### 2.2. Mathematical Model

Continuity equation

The form of continuity equation in cylindrical coordinate system is as follows:

$$\nabla \cdot \mathbf{u} = 0 \tag{1}$$

Equation (1) applies to incompressible fluids, i.e., $\rho\,(r, z, t) = c$, where $t$ is time and $z$ and $r$ are axial and radial directions.

### 2.2.1. Momentum Equation

The form of momentum equation in cylindrical coordinate system is as follows:

$$\rho \frac{\partial u}{\partial t} + (\rho \mathbf{u} \cdot \nabla)\mathbf{u} = -\nabla p + \nabla \cdot \tau - \rho \beta\,(T - T_{\text{ref}})g \tag{2}$$

where

$$\tau = \mu \left[ \left( \nabla \mathbf{u} + \nabla \mathbf{u}^T \right) \right] \tag{3}$$

where $p$ is the static pressure, $\tau$ is the stress tensor, $\rho \beta\,(T - T_{\text{ref}})\,g$ is the volume force, and $\mu$ is dynamic viscosity.

### 2.2.2. Heat Transfer Equation

The temperature distribution is calculated by the diffusion/convection equation as follows:

$$\rho C_{\text{p}} \frac{\partial T}{\partial t} + \rho C_{\text{p}} \mathbf{u} \cdot \nabla T = \nabla \cdot (\lambda \nabla T) \tag{4}$$

where $C_{\text{p}}$ is the specific heat capacity at constant pressure and $\lambda$ is thermal conductivity.

### 2.2.3. Turbulence Model Equation

The standard turbulence model is adopted:

$$\frac{\partial}{\partial t}\,(\rho k) + \nabla \cdot (\rho \mathbf{u} k) = \nabla \cdot \left[ \left( \mu + \frac{\mu_t}{\sigma_k} \right) \nabla k \right] + G_{\text{k}} + G_{\text{b}} - \rho \varepsilon \tag{5}$$

$$\frac{\partial}{\partial t}\,(\rho \varepsilon) + \nabla \cdot (\rho \mathbf{u} \varepsilon) = \nabla \cdot (\Gamma_\varepsilon \nabla \varepsilon) + \rho \frac{\varepsilon}{k}\,(c_1 G - c_2 \varepsilon) \tag{6}$$

where $G_k$ is the turbulent kinetic energy generated by velocity gradient, $G_b$ is the turbulent kinetic energy generated by buoyancy, and $k$ and $\varepsilon$ are fluid turbulent kinetic energy and turbulent dissipation rate, respectively.

*2.3. Numerical Calculation Method*

2.3.1. Boundary Conditions and Solver Settings

The structure of a single storage tank is simple, so the structured quadrilateral element grid and the axisymmetric plane model are adopted. Fluent's two-dimensional double precision solver is employed in the calculation. The coupling, implicit and two-dimensional axisymmetric method is used in the solver. The physical model adopts unsteady, heat transfer and turbulence model ($Re = \frac{\rho u_{in} D}{\mu} = \frac{974.03 \text{ kg/m}^3 \cdot 4.567 \times 10^{-4} \text{ m/s} \cdot 22 \text{ m}}{370.85 \text{ }\mu\text{Pa} \cdot \text{s}} = 26,389 > 12,000$). Gravity is considered in operating conditions. The second-order upwind scheme is used for discretization. The inlet and outlet boundary conditions are velocity inlet and outflow, respectively, and the wall boundary condition is adiabatic. During charging process, the initial temperature in the storage tank is $T_c = 328$ K; During the discharging process, the initial temperature in the storage tank is $T_d = 371$ K. PISO algorithm is used to realize the coupling calculation of pressure and velocity. In the solver, the relaxation factor of energy is set to 0.8 and the relaxation factor of momentum is set to 0.7.

2.3.2. Physical Parameters of Working Fluid

The working fluid is water under normal pressure and 328~371 K, and its main physical parameters are as follows:

According to the physical property parameter data of water, the following fitting correlations have been achieved to calculate the density, dynamic viscosity, and thermal conductivity of working fluid during the range of the given temperature.

$$\rho \text{ (kg} \cdot \text{m}^{-3}) = 866.19608 + 1.22008T - 0.00261T^2 \tag{7}$$

$$\mu \text{ (Pa} \cdot \text{s)} = \frac{1.779 \times 10^{-5}}{8.276269 - 0.086986T + 0.000221T^2} \tag{8}$$

$$\lambda \text{ (W/m} \cdot \text{K)} = -46.99 + 24.106 \times (1 - e^{-\frac{T}{48.09069}}) + 23.58177 \times (1 - e^{-\frac{T}{48.10424}}) \tag{9}$$

Because there is just a little variation for the specific heat capacity at the given constant pressure within the temperature range in our cases, it is taken as a constant as the following.

$$C_p = 4.19805 \times 10^3 \text{ J/(kg} \cdot \text{K)} \tag{10}$$

2.3.3. Grid Independence Verification

In order to select an efficient and accurate combination of grid and time step, the flow rate of 625 m$^3$/h and the height-to-diameter ratio of 1.2 are taken as an example to verify the independence of the grid model.

From Figure 3, it is observed that the simulation results are almost consistent under the designed grid densities and time steps. Therefore, it is considered that the grid density corresponding to the black solid line has met the requirement of obtaining independent solutions and meets the grid independence.

2.3.4. Validation

To verify the accuracy of the model, numerical simulations are conducted under the condition of $Q = 0.5$ m$^3$/h, $T_H = 356$ K, and $T_L = 323$ K in the charging process, and the results are compared with the experimental data of Wang et al. [14]. The axial temperature distribution at each time of the storage tank is shown in Figure 4.

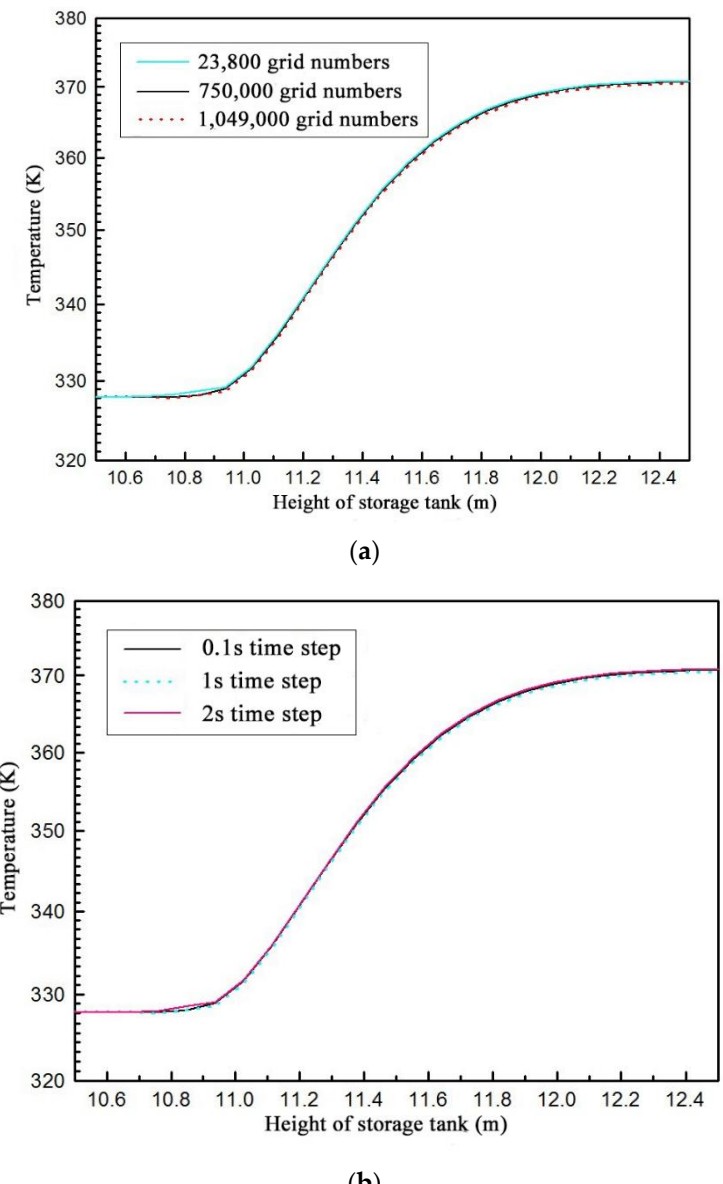

**Figure 3. (a)** Temperature diagram of three groups of tank axes with different grid numbers. **(b)** Temperature diagram of three groups of tank axes with different time steps.

It can be seen in Figure 4 that the main errors between numerical results and experimental results are concentrated on the top and bottom of the storage tank. There are two main reasons. During the experiment, the inlet and outlet at the top and bottom of the tank have a large disturbance to the fluid temperature in the tank, so the error is large; During the simulation, the irregular geometry of the top and bottom of the tank is simplified, causing a certain difference between numerical results and experimental results in the middle and late stage of charging. Through the comparison of numerical results and experimental results, the maximum relative error is about 1.3%. Considering the causes of the above errors, it can be considered that the numerical results in this paper are rather accurate.

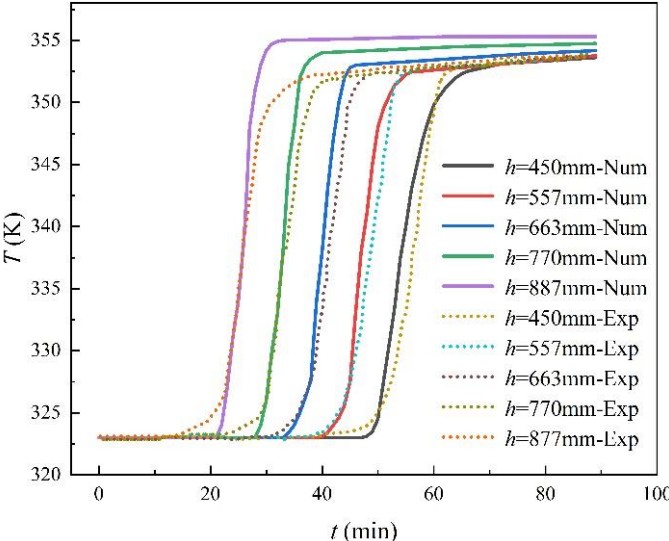

**Figure 4.** The axial temperature distribution at each time of the storage tank (Comparison of numerical results and experimental results).

## 3. Results

In this paper, the critical temperature of the upper interface of the thermocline is selected as 370.5 K and 368 K, respectively, in the charging and discharging processes, while the critical temperature of the lower interface of the thermocline is selected as 328.5 K and 331 K respectively in charging and discharging processes. Because the basic trend of the two is consistent, $\pm 0.5$ K is taken as an example.

The existence of thermocline will reduce the high-temperature water that can be used for effective heating in the storage tank, then reduce the performance of the TES system. The thicker the thermocline is, the worse the performance is. Therefore, taking the thermocline thickness as a reference, the following formula can be used to calculate the performance evaluation index of the TES system:

$$\kappa = 1 - \frac{\delta}{H} \tag{11}$$

where $\delta$ is the thermocline thickness and $H$ is the total height of the storage tank.

The larger the performance evaluation index is, the higher the efficiency of the TES system is.

### 3.1. The Effect of Time

In the simulation about time, the geometric model is as follows: the height of storage tank $H$ is 26.3688 m, the diameter $D$ is 21.974 m, inlet velocity $u_{in}$ is 0.00045779 m/s; during charging process, inlet temperature $T_{in}$ = 98 °C = 371 K, outlet temperature $T_{out}$ = 55 °C = 328 K; during discharging process, inlet temperature $T_{in}$ = 55 °C = 328 K, outlet temperature $T_{out}$ = 98 °C = 371 K. Under the above condition, the charging and discharging processes of the TES system are simulated respectively, and the temperature distribution in the storage tank and its variation with time are obtained, as presented in Figure 5.

Figure 6 illustrates the variation of thermocline thickness in the storage tank during the charging and discharging processes. From Figures 5 and 6, it is observed that the thermocline in the storage tank gradually thickens with the charging or discharging time, but the growth of thermocline thickness decelerates gradually. It is seen that the thermal disturbance caused by the heat transfer in the axial direction of the thermocline with a large temperature gradient deepens with the processes of charging and discharging, and then gradually weakens and tends to be stable with time.

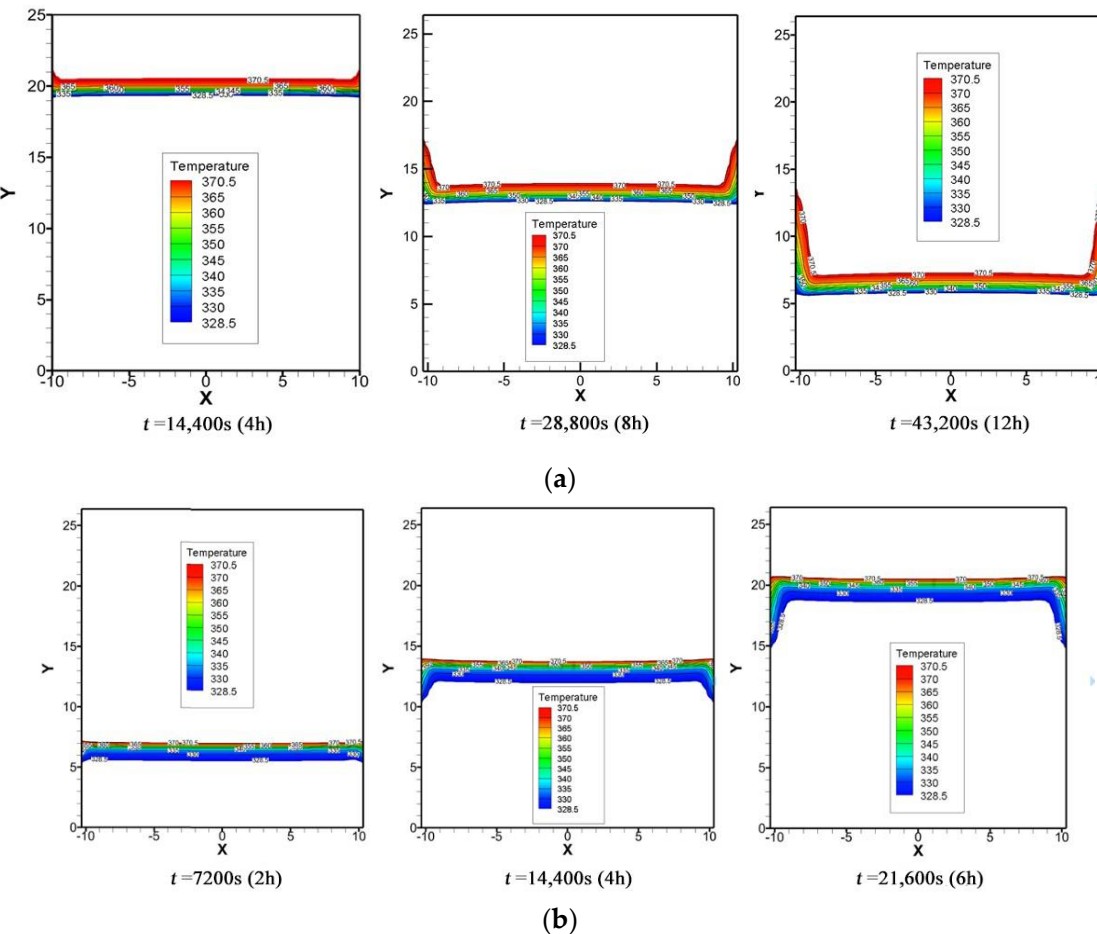

**Figure 5.** The variation of temperature distribution in the storage tank ((**a**)-charging, (**b**)-discharging).

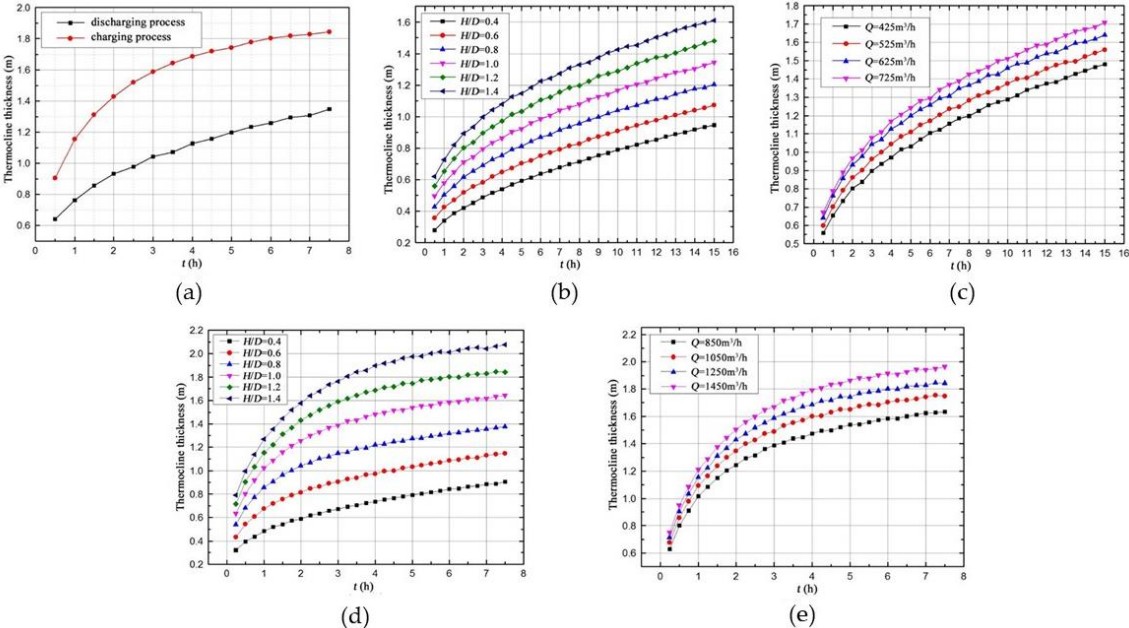

**Figure 6.** The variation of thermocline thickness in the storage tank during charging and discharging process: (**a**) comparison of charging and discharging, (**b**) comparison of height to diameter ratios (charging), (**c**) comparison of flow rates (charging), (**d**) comparison of height to diameter ratios (discharging), (**e**) comparison of flow rates (discharging).

During the charging and discharging processes, the thermocline thickness in the storage tank is increasing, so the charging and discharging efficiency of the TES system is decreasing. In practice, the growth of thermocline thickness should be as gentle as possible with time to maintain the efficient and stable running of the TES system.

### 3.2. The Effect of Flow Velocity

In order to keep the flow disturbance in the storage tank within a certain range and maintain the good working performance of the TES system, the inlet flow rate is usually set at a low level during the charging and discharging processes. However, the inlet velocity is related to the flow and heat transfer parameters such as inlet flow rate and Reynolds number in the storage tank. The variation within its reasonable range may cause the change in charging and discharging characteristics of the TES system. It's necessary to investigate the influence of the change of inlet velocity on the thermocline in the storage tank, for selecting the best inlet velocity to improve the efficiency of the TES system.

In the simulation of flow rate, other parameters remain unchanged. The inlet flow velocity in the charging process is $4.0\sim4.8 \times 10^{-3}$ m/s, and the inlet flow velocity in the discharging process is $8.0\sim9.6 \times 10^{-3}$ m/s. Figure 7 presents the variation of thermocline thickness with flow rate during charging and discharging processes at different times.

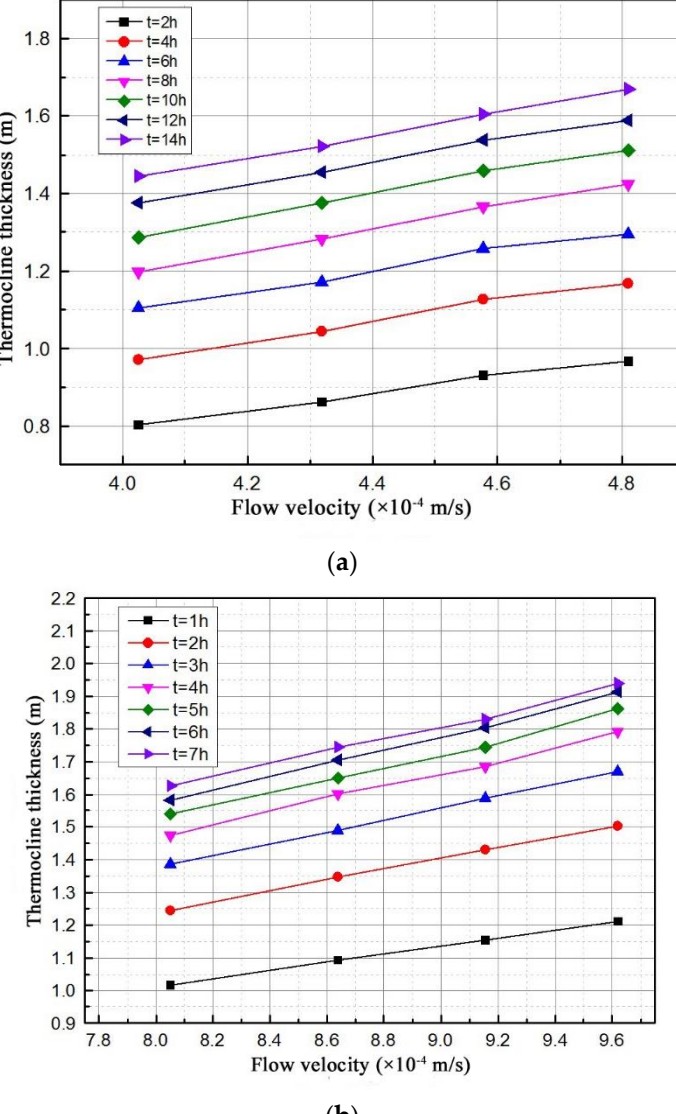

**Figure 7.** The variation of thermocline thickness with flow velocity ((**a**)-charging, (**b**)-discharging).

It can be seen in Figure 7 that the thermocline thickness increases with the increase of the inlet flow velocity, whether charging or discharging. When the inlet velocity increases, the average velocity in the storage tank increases, and the Reynolds number of turbulent flow in the storage tank increases accordingly. The increase of Re will lead to the increase of Nu. It means that the convective heat transfer of water in the storage tank is enhanced, which enhances the thermal disturbance in the storage tank and thickens the thermocline.

Under different inlet flow velocities, the thermocline thickness in the storage tank increases with the increase of inlet flow velocity during charging and discharging processes, but the system performance index defined by Equation (11) combined with Table 1 shows that the performance of the TES system has almost no change. This is because this study is based on the change of velocity under different flow rates. That is, when the velocity increases, the flow rate increases accordingly. When the charging time and height to diameter ratio are certain, it means that the diameter and height of the tank body increase. In this way, it is found that the ratio of the thermocline thickness to the height of the tank body and the evaluation index are approximately unchanged.

**Table 1.** The thermocline thickness and performance index value of TES system in charging process at different flow rates (*t* = 28,800 s).

| Inlet Velocity $u_{in}/\times 10^{-4}$ m·s$^{-1}$ | Height to Diameter Ratio: 0.4 | | Height to Diameter Ratio: 1.2 | |
|---|---|---|---|---|
| | Thermocline Thickness/m | Performance Index | Thermocline Thickness/m | Performance Index |
| 4.025 | 0.713 | 0.936 | 1.197 | 0.948 |
| 4.319 | 0.741 | 0.938 | 1.283 | 0.948 |
| 4.577 | 0.768 | 0.939 | 1.366 | 0.948 |
| 4.810 | 0.782 | 0.941 | 1.425 | 0.949 |

### 3.3. The Effect of Height-to-Diameter Ratio

The other two macro structural parameters of the storage tank, namely diameter, and height will also have an impact on the flow and heat transfer characteristics in the tank. The height of the tank affects the development of flow in the tank, while the diameter of the tank affects the macro characteristic parameters of flow in the tank such as the Reynolds number. In order to comprehensively consider the effects of the two, the structural parameter height to diameter ratio $H/D$ is proposed to investigate the charging and discharging characteristics of the TES system under different structures of the storage tank. The models of the thermocline storage tank with height to diameter ratios $H/D$ = 0.4/0.5/0.6/0.7/0.8/0.9/1.0/1.1/1.2/1.3/1.4 are established respectively.

On the one hand, the increase of height-to-diameter ratio $H/D$ means that the ratio $D/H$ of diameter $D$ to total height $H$ of the storage tank decreases, causing a decrease in average Nu and thermal disturbance in the tank. On the other hand, on the premise that the total volume of the storage tank remains unchanged, the increase of the height to diameter ratio means that the diameter of the storage tank decreases. At this time, the flow rate remains unchanged, leading to an increase in the flow velocity and the Reynolds number in the storage tank, which will enhance the disturbance in the tank.

Figure 8 shows the variation of the thermocline thickness with the height to diameter ratio at different times in the charging and discharging processes. It can be clearly seen that when the height to diameter ratio of the storage tank rises, the thermocline thickness in the tank increases. However, the performance index of the TES system continues to increase, indicating that the charging and discharging efficiency continues to improve. It is because the increase of the height to diameter ratio leads to the increase of the total height, and the increase of the total height is much greater than that of the thermocline thickness. The share of the thermocline thickness in the total height drops with the rise of the height to diameter ratio, so the efficiency of the TES system is improved. In addition, it can be seen in Figure 9 that the thermocline in discharging process is thicker than that in charging process

with the same time. Obviously, the larger the height to diameter ratio $H/D$, the thicker the thermocline will be, but as illustrated in Figure 10 that the performance of the TES system will be improved accordingly.

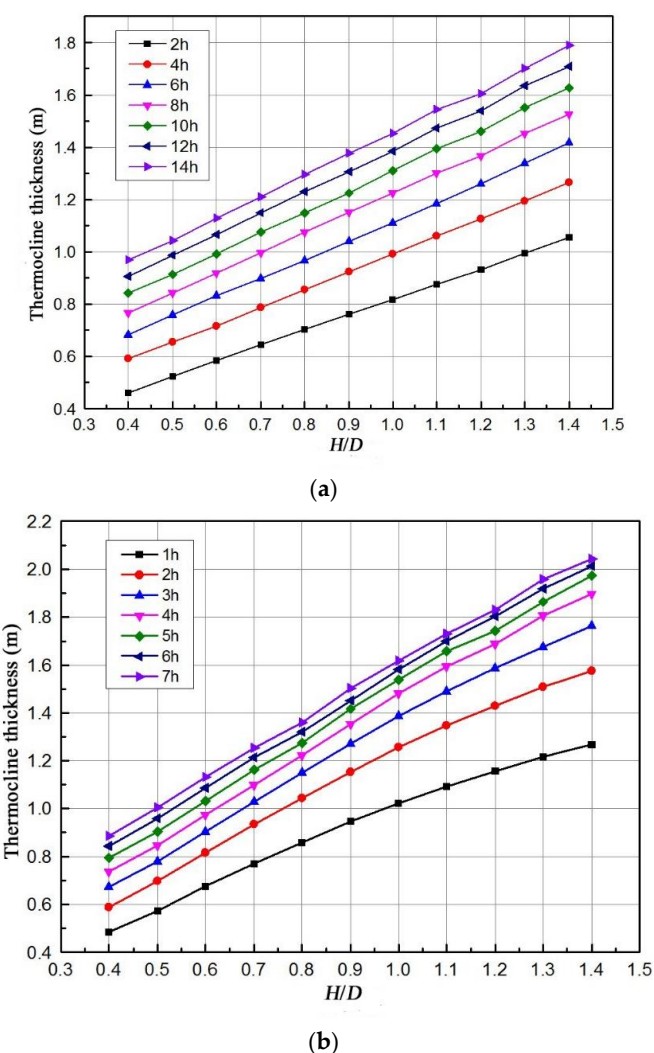

(**a**)

(**b**)

**Figure 8.** The variation of thermocline thickness with height to diameter ratio ((**a**)-charging, (**b**)-discharging).

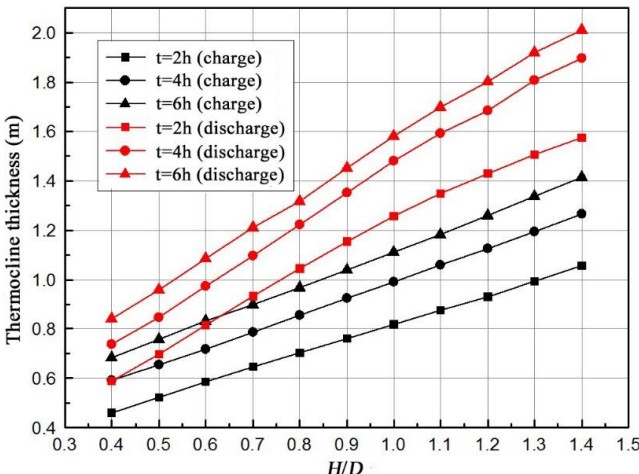

**Figure 9.** Comparison of thermocline thickness in charging and discharging processes.

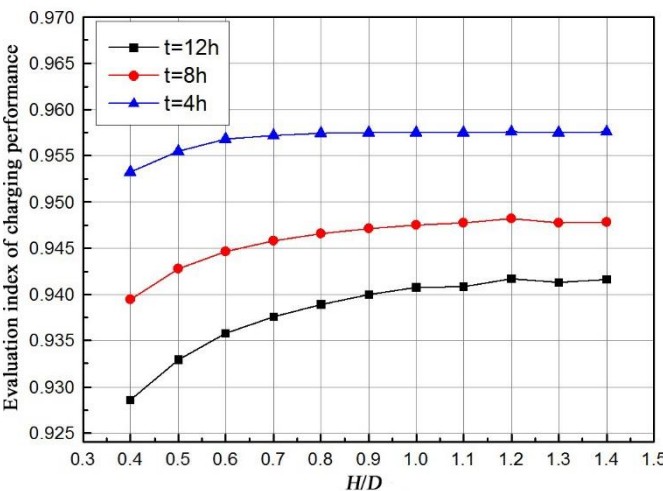

**Figure 10.** The variation of performance index of TES system with height to diameter ratio in the charging process.

## 4. Conclusions

In this study, a two-dimensional flow and heat transfer model of a single-tank thermal energy storage system is established, and the effects of time, flow velocity, and height to diameter ratio on thermocline thickness are investigated respectively. Taking the thermocline thickness as a reference, a performance evaluation index is proposed to evaluate the thermal performance of a single-tank thermal energy storage system. The conclusions are as follows.

1. During the charging and discharging processes, the thermocline thickness in the storage tank is increasing while the charging and discharging efficiency of the TES system is decreasing.
2. The thermocline thickness in the storage tank increases with the increase of inlet flow velocity during charging and discharging processes, but the system performance index has almost no change.
3. When the height to diameter ratio of the storage tank rises, the thermocline thickness in the tank increases, but the performance index of the TES system continues to increase.
4. In practice, the growth of thermocline thickness should be as gentle as possible with time to maintain the efficient and stable running of the TES system. In this study, it seems that inlet flow velocity has little influence on the thermal performance of the system. A high height to diameter ratio is beneficial to improve the charging and discharging efficiency.

**Author Contributions:** Conceptualization, C.Z. and Y.W.; methodology, Y.W. and Z.D.; software, Z.D.; validation, J.Z., Z.D. and Y.W.; formal analysis, J.Z. and Y.W.; investigation, Z.D. and J.Z.; resources, Z.D.; data curation, J.Z. and Y.W.; writing—original draft preparation, J.Z.; writing—review and editing, Y.W. and J.Z.; visualization, C.Z.; supervision, Y.W.; project administration, P.S., J.W. and Z.W.; funding acquisition, C.Z. All authors have read and agreed to the published version of the manuscript.

**Funding:** This research was funded by Science Research Project of State Grid Shaanxi Electric Power Company (Research and application of high efficiency photovoltaic—photothermal—energy storage coupling clean energy supply Technology, No. 5226KY22000Z).

**Institutional Review Board Statement:** Not applicable.

**Informed Consent Statement:** Not applicable.

**Data Availability Statement:** Not applicable.

**Acknowledgments:** This work is supported by Science Research Project of State Grid Shaanxi Electric Power Company (Research and application of high efficiency photovoltaic—photothermal—energy storage coupling clean energy supply Technology, No. 5226KY22000Z).

**Conflicts of Interest:** The authors declare no conflict of interest.

## Nomenclature

| | |
|---|---|
| $c_{\mathrm{p}}$ | specific heat capacity at constant pressure/$\mathrm{J \cdot kg^{-1} \cdot K^{-1}}$ |
| CFD | Computational Fluid Dynamics |
| CHP | Combined Heat and Power |
| CSP | Concentrating Solar Power |
| $D$ | diameter of storage tank/m |
| $g$ | acceleration of gravity/$\mathrm{m \cdot s^{-2}}$ |
| $G_{\mathrm{b}}$ | turbulent kinetic energy generated by buoyancy/$\mathrm{kg \cdot m^{-1} \cdot s^{-3}}$ |
| $G_{\mathrm{k}}$ | turbulent kinetic energy generated by velocity gradient/$\mathrm{kg \cdot m^{-1} \cdot s^{-3}}$ |
| $H$ | height of storage tank/m |
| HTF | Heat Transfer Fluid |
| $k$ | fluid turbulent kinetic energy/$\mathrm{m^2 \cdot s^{-2}}$ |
| $p$ | pressure/$\mathrm{kg \cdot m^{-1} \cdot s^{-2}}$ |
| $Q$ | rate of flow/$\mathrm{m^3 \cdot h^{-1}}$ |
| $R$ | radial direction/m |
| SMT | Single Medium Thermocline |
| $t$ | time/s |
| $T_{\mathrm{c}}$ | the initial temperature in the storage tank during charging process/K |
| $T_{\mathrm{d}}$ | the initial temperature in the storage tank during discharging process/K |
| $T_{\mathrm{H}}$ | highest temperature of thermocline/K |
| $T_{\mathrm{in}}$ | inlet temperature/K |
| $T_{\mathrm{L}}$ | lowest temperature of thermocline/K |
| $T_{\mathrm{out}}$ | outlet temperature/K |
| $T_{\mathrm{ref}}$ | reference temperature/K |
| $T$ | temperature/K |
| TES | Thermal Energy Storage |
| $u_{\mathrm{in}}$ | inlet velocity/$\mathrm{m \cdot s^{-1}}$ |
| $\mathbf{u}$ | velocity vector |
| $z$ | axial direction/m |
| $\delta$ | thermocline thickness/m |
| $\varepsilon$ | turbulent dissipation rate/$\mathrm{m^2 \cdot s^{-3}}$ |
| $\kappa$ | performance evaluation index of the TES system |
| $\lambda$ | thermal conductivity/$\mathrm{W \cdot m^{-1} \cdot K^{-1}}$ |
| $\mu$ | dynamic viscosity/$\mathrm{kg \cdot m^{-1} \cdot s^{-1}}$ |
| $\rho$ | density/$\mathrm{kg \cdot m^{-3}}$ |
| $\tau$ | stress tensor |

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
