# Peer review of "Study on Thermal Performance of Single-Tank Thermal Energy Storage System with Thermocline in Solar Thermal Utilization"

_applsci, doi:10.3390/app12083908_

Round 1
Reviewer 1 Report
The research work is conducted and presented well. There are some comments (27 Nos) as annotated in the pdf file of the manuscript. Authors are invited to revise the manuscript addressing those comments diligently to further improve the quality of the paper. Besides, a schematic of the complete system needs to be added to improve the understanding and application of the thermal energy storage tank.
A section for symbol is to be added.
Citation of the references are to be checked and corrected.

Reviewer 2 Report
In this work, the authors reported the thermal performance of a single-tank thermal energy storage system with thermocline in solar thermal utilization. The manuscript was well organized. The Introduction part was clearly written and the objectives were clearly stated in the introduction. Some points must be provided and/or revised before consideration for publication.
- The highlight values obtained in this study must be mentioned in the Abstract.
- Although the Introduction was well written and organized, the novelty of this work should be clearly mentioned in the introduction.
- Figures 5a-e should be bundled in a single figure.
- Please check the decimal number in Table 1.
- Please check the second term on the righ hand side of eq. (6).
- Please check, Figure 2a is missing.
Reviewer 3 Report
This work reports on the analytical approach solution from 2D-flow and heat transfer model of single-tank thermal energy storage system using water as heat transfer fluid. The authors simulated the thermal behaviour in the cylindrical storage tank with water and revealed that the thermocline in the tank expands continually but its extension slows down with time.
The authors also presented that the higher the ratio of height to diameter of the storage tank, the higher the thermal performance index.
This manuscript seems to be useful in the field of design and operation of the single-tank thermal energy storage system for effective use of solar energy.
I think this manuscript deserves publication after minor revision.
Comment:
p.4, line 152
950, 1050, 1250 and 1450m3/h ….
‘950’ may be fixed to ‘850’.
p.5, line 218-219
Tin = 328k
Tin = 371k
‘k’ may be fixed to ‘K’.
p.9, line 303
Figure 5b. caption
‘The variation of the thermocline thickness in the storage tank during charging and discharging processes (comparison of charging and discharging).’ b) The variation of thermocline thickness in the storage tank during charging process (comparison of height to diameter ratios)
The first sentence may be a caption for Figure 5a and it should be deleted for Figure 5b.
p.10, line 311
Figure 5d. caption
(comparison of height to diameter ratios)
‘diameter ratios’ may be fixed to ‘flow rates’.
p.11, line 315
Figure 5e. caption
(comparison of height to flow rates)
‘flow rates’ may be fixed to ‘diameter ratios’
Reviewer 4 Report
The manuscript is very well written, understandable and interesting. It deals with the modelling, and it is also partly compared with the experiment presented in previously published paper. It is a pity that it is not compared with more experiments from different authors or experiments carried out under different conditions. However, this does not diminish the importance of work. I have only a few formal comments on the manuscript:
It would be useful to summarize the list of abbreviations and symbols
Figure 2a is missing
In the text, the Kelvin temperature unit is written as “k” (e.g. page 5, lines 218,219
Equations and formulas should be written in the same font size as the rest of the text, otherwise it disturbs the reader very much
Include references to equations (7) - (9)
Round 2
Reviewer 1 Report
The authors are appreciated for making all the necessary corrections which would improve the quality of the manuscript.